# An Update on Antibodies to Necleosome Components as Biomarkers of Sistemic Lupus Erythematosus and of Lupus Flares

**DOI:** 10.3390/ijms20225799

**Published:** 2019-11-18

**Authors:** Gian Marco Ghiggeri, Matteo D’Alessandro, Domenico Bartolomeo, Maria Ludovica Degl’Innocenti, Alberto Magnasco, Francesca Lugani, Marco Prunotto, Maurizio Bruschi

**Affiliations:** 1Division of Nephrology, Dialysis and Transplantation, Istituto G. Gaslini, Largo G. Gaslini 5, 16147 Genoa, Italydomenico.bartolomeo@libero.it (D.B.); albertomagnasco@gaslini.org (A.M.); 2Laboratory of Molecular Nephrology, Scientific Institute for Research and Health Care, IRCCS IstitutoGianninaGaslini, 16147 Genoa, Italy; francescalugani@ospedale-gaslini.ge.it (F.L.); mauriziobruschi@gaslini.org (M.B.); 3School of Pharmaceutical Sciences, University of Geneva, 1211 Geneva, Switzerland; marco.prunotto@unige.ch; 4Fondazione per le MalattieRenalinel Bambino, 16100 Genoa, Italy

**Keywords:** systemic lupus erythematosus, biomarkers, ant-dsDNA antibodies, anti-nucleosomeanti-Histone antibodies

## Abstract

Systemic lupus erythematosus (SLE) is an autoimmune disease with variable clinical expression. It is a potentially devastating condition affecting mostly women and leading to clinically unpredictable outcomes. Remission and flares may, in fact, alternate over time and a mild involvement limited to few articular sites may be followed by severe and widespread organ damage. SLE is the prototype of any autoimmune condition and has, for this reason, attracted the interest of basic immunologists. Therapies have evolved over time and clinical prognosis has, in parallel, been improved. What clinicians still lack is the possibility to use biomarkers of the disease as predictors of outcome and, in this area, several studies are trying to find solutions. Circulating autoantibodies are clearly a milestone of clinical research and the concrete possibility is to integrate, in the future, classical markers of activation (like C3) with target organ autoantibodies. Anti-dsDNA antibodies represent a basic point in any predictive attempt in SLE and should be considered the benchmark for any innovative proposal in the wide field of target organ pathologies related to SLE. DNA is part of the nucleosome that is the basic unit of chromatin. It consists of DNA wrapped around a histone octamer made of 2 copies each of Histone 2A, 2B, 3, and 4. The nucleosome has a plastic organization that varies over time and has the potential to stimulate the formation of antibodies directed to the whole structure (anti-nucleosome) or its parts (anti-dsDNA and anti-Histones). Here, we present an updated review of the literature on antibodies directed to the nucleosome and the nucleosome constituents, i.e., DNA and Histones. Wetriedto merge the data first published more than twenty years ago with more recent results to create a balanced bridge between old dogma and more recent research that could serve as a stimulus to reconsider mechanisms for SLE. The formation of large networks would provide the chance of studying large cohorts of patients and confirm what already presented in small sample size during the last years.

## 1. Anti-dsDNA

Anti-dsDNA antibodies were discovered in 1957 [1,2,3,4] and represent the first autoantibodies described in SLE. They have been unanimously considered the real marker of SLE for over 50 years and their presence in blood has been included in the ACR and SLICC criteria [5,6] and in the more recent 2019 EULAR/ACR classification [7]. Anti-dsDNA antibodies have a prevalence between 60 and 90% of SLE patients with a specificity that is variable within other rheumatologic conditions, thus opening a“de facto” discussion on overlaps of clinical significance.

### 1.1. DNA Structure and Mechanisms for Anti-dsDNA Formation

Mammalian dsDNA comprises B-DNA and Z-DNA that are denoted as right- and left-handed specular forms of the same double helix [8]. They have different immunogenic characteristics: Z-DNA is a potent and stable inducer of antibodies [9], B-DNA may, instead, be immunogenic only under specific conditions [10,11]. The specificity of anti-dsDNA antibodies for Z-DNA or B-DNA and for other nucleosome structures, including histones and ssDNA, has not been clearly defined and provides further support to discussion.

Basic research demonstrated the complexity of anti-dsDNA antibodies formation; see for example the recent review by Rekvig [12]. In this context, under appropriate co-stimulatory T_H_ cell signal, the MCH class II complex of B-cells processes nucleosome components based on molecules exposed on the surface at any given moment. The fluid nature of chromatin in nucleosome allows the transitory exposure of different antigenic sites [13,14]; in some cases, the exposed molecules are peptides of viral or bacterial origin that determine the production of protective antibodies against the infectious trigger [10,15]; in case of SLE, the stimulatory peptides are autologous components and determine an autoimmune response [16,17]. B-DNA, for example, may expose arginines that are recognized by particular cationic epitopes of the immunoglobulin heavy variable region of anti-DNA (VH-CDR3) [18]. All these processes are transitory in nature [19,20,21,22] and lead to the production of two different types of anti-dsDNA antibody that has different avidity for the antigen: the former, with low affinity, is biomarker of disease flares because of infectious triggers; the later, with high affinity, is biomarker of true flares because of the autoimmune process [23]. A second main question is about the linear or bent forms of DNA utilized in anti-DNA tests since both types are structural components of nucleosome, one (bent DNA) is the DNA wrapped around histones, the other (linear DNA) is the DNA of the linker region.

The possibility of different haptens and triggers, the existence of different antibodies with low and high avidity and the variegate nature of nucleosome components which are variably exposed, support a generic concept of non-specificity of anti-dsDNA antibodies that rebounds in clinical practice and limit their use as classificatory and predictive parameter.

### 1.2. Technical Aspects Related to Anti-dsDNA Determination

A variety of techniques have been utilized over the years to quantify, or in some cases only identify, circulating anti-dsDNA (for a review see [24]). Currently, the enzyme-linked immune-adsorption (ELISA) and immune-fluorescence (*Crithidia luciliae*) are the most commonly utilized techniques while the radioimmunoassay Farr, that has been very popular in the past, is nowadays less utilized for disadvantages linked to radioactivity. There are important differences among all these techniques that hint in some way the comparison of results from different studies; the Farr assay detects principally high avidity antibodies but cannot distinguish between IgG and IgM, the immune-fluorescence assay with *Crithidia luciliae* is highly specific for high affinity antibodies and probably for bent DNA but it is less sensitive and only sometimes quantitative; finally, the ELISAs are sensitive but less specific since determine low and high affinity antibodies and also capture both linear and bent-DNA. Studies performed between 1970 and 1990 compared performances of different techniques in population with SLE and indicated that overall, there is statistical correlation [25,26,27,28]; however, discrepancies have been reported within different methods and also within different commercial kits utilizing the same technique (i.e., ELISA, IF and Farr) [29,30]. In more recent years, only few studies have compared different assays for anti-dsDNA in large cohorts of patients [31,32,33]. Results have confirmed preceding analysis overall indicating the different specificity and variability of different anti-DNA assays which is a limit for a clear comparison among studies that utilized different assay.

### 1.3. Anti-dsDNA Antibodies Are Markers of SLE

The above descriptions of the nucleosome structure and of the mechanisms involved in antibodies production versus dsDNA and versus other constituents of nucleosome, represents a necessary premise to a discussion on their clinical significance. As already reported, anti-dsDNA antibodies had been included as diagnostic criterion in ACR and SLICC [5,6] and, now, in the 2019 EULAR/ACR classification [7]. In the later report by EULAR/ACR, anti-dsDNA positivity is limited to those assays with 90% specificity; stressing, therefore, the key importance of the method utilized for anti-dsDNA determination. The test should be done in patients with ANA positivity (that is the unique inclusion criterion) and when anti-dsDNA are present they are powdered 6 in a scale that classifies as SLE any patient accumulating more than 10 points [7]. Previous reports indicated that anti-dsDNA positivity in patients negative for ANA is <1% [34,35] thus underlining the concept that circulating anti-dsDNA dosage must be reserved to patients with a positive ANA test.

Most data of the literature and the meta-analysis agree that anti-dsDNA testing is very useful for the diagnosis of SLE with a sensitivity of 57.3% and a specificity of 97.4% and with a very high positive likelihood ratio >16 [36] that means high probability of SLE in case of a positive test. Sensitivity is, instead, specular to a low negative likelihood ratio of 0.49 which implies that a negative anti-dsDNA does not exclude per se the presence of SLE. On the other hand, circulating anti-dsDNA antibodies have been described in other rheumatologic and immunologic conditions with a frequency that is less than 5% implying that performing an anti-dsDNA testing is useful for a confirmation of SLE in ANA positive patients for whom the disease has been suspected on clinical basis.

### 1.4. Association with Lupus Flares and Prognosis

There is a wide literature considering anti-dsDNA actively implicated in causing acute disease manifestations of lupus and more in general in the long-term outcome. For validating the association between anti-dsDNA and lupus flares it is required to demonstrate that one or more symptoms can be transferred or reproduced in experimental models by infusion of anti-dsDNA antibodies; a second validation criterion is the clear proof of a clinical association with symptoms. Both aspects are reviewed here and in the following section in relation to lupus nephritis.

In general, lupus flares reflect the release of cytokines by plasmacytoid dendritic cells [37,38], in particular of type 1 interferon, that produce systemic symptoms such as fever, fatigue, malaise. The same symptoms may also occur as reaction to bacterial or viral DNA and are a basic defense mechanism. Cells are stimulated to release cytokines when DNA is introduced inside them by immune-complexes containing dsDNA [39] that are formed in circulation [40] indicating that the levels of antibodies should have a role in symptoms. So far, there is not a clear demonstration that levels of circulating anti-dsDNA antibodies correlate with lupus activity [41,42] also because in clinical trials, the presence of circulating anti-dsDNA is one of the SLE activity score criteria [43]. In the SLEDAI score, for example, the presence of circulating anti-dsDNA is valued 2 where an increase of 3 points is considered a mild flare. In spite of this caveat, there are many reports in the literature considering the clinical association between anti-dsDNA and disease activity. These aspects have been debated in their “Guidelines for Immunologic Laboratory Testing in Rheumatic Diseases” by Kavanaugh and coll. [36] who found, by reviewing the literature *ante* 2002, 20 papers which were graded as informative. Results were variable, probably deriving from patients heterogeneity and largely depending on the antibody levels; while the weighted mean positive likelihood ratio was 4.14, that is low and suggests a small effect; high titers of antibodies were more strongly associated with lupus activity [36]. This means that the mere presence of anti-dsDNA antibodies in a given patient with SLE has a limited impact in predicting lupus activity and outcome; high levels should, instead, be considered an important predictive element. On the other side, the weighted negative likelihood ratio for the same association was 0.51 indicating that a negative test does not exclude the presence of active SLE. Gensous and coll. [44] made a systematic review of the literature appeared in the years *ante* 2014 and retrieved 69 studies from 4668 records that were considered of interest and 28 reported anti-dsDNA; with the exception of 4 papers; results on anti-dsDNA were obtained with only one technique with a predominance of the Farr assay. Eighteen studies analyzed 2881 patients overall and were in favor of a predictive role of anti-dsDNA for SLE exacerbations; in the major cohort that included 562 patients, anti-dsDNA titers >200 IU/mL at baseline were independent predictors of moderate-to-severe flares after 4 years [45]. Increased levels of anti-dsDNA levels at baseline were an independent predictor of hematologic flares as well [46]. When the level of anti-dsDNA was evaluated during the course of the disease, nine studies out of fifteen reported an association [42,47,48,49], whereas six failed to find any one [50,51,52]. Overall, the conclusion of the review is that, in spite of some inconsistencies, high levels of circulating anti-dsDNA antibodies do predict SLE flares and can be utilized as reliable biomarker.

### 1.5. Association with Lupus Nephritis

Lupus nephritis (LN) is the most frequent complication of SLE occurring in almost 50% of the patients. It is a serious condition that needs a rapid clinical assessment and therapy [1]. LN is also the most studied organ localization of SLE and anti-dsDNA have been considered for a long time an important mechanism of the pathology. The following considerations support a direct role of anti-dsDNA in LN pathogenesis: (1) anti-dsDNA (along with several other autoantibodies) can be eluted from the renal glomeruli of patients with LN [53,54]. One interesting characteristic of eluted antibodies is their IgG2 and IgG3 isotypes [55,56,57] that suggests specific TLRs 8-9 drive the isotype switch (see below); (2) in glomeruli, anti-dsDNA antibodies bind chromatin fragments in mesangial matrix (in early lesions) and in glomerular basement membrane (GBM) (in more advanced stages of nephritis) [53,58,59]; (3) injection of anti-dsDNA in mice, bind GBM and cause deposition of chromatin-IgG complex [12,60]; (4) anti-dsDNA can bind also non nucleosome proteins such as annexin A2 [61,62] that is a protein involved in activation of inflammatory pathways and ligates C1q [63].

A clinical association of anti-dsDNA with LN has been considered from many years. In their “Guidelines for Immunologic Laboratory Testing in Rheumatic Diseases” Kavanaugh and coll. [36] made a survey analysis of studies published *ante* 2002 and reported a low positive likelihood ratio of 1.7, indicating that anti-dsDNA has a limited value to make any distinction in SLE patients with and without nephritis. The high percent of positivity (that in some studies reaches 90%) for anti-dsDNA antibodies in SLE patients with and without nephritis could, in some way, explain why it is difficult to find any association between the two parameters and predicts that only studies enrolling very numerous cohorts of patients could show or deny the association. Actually, the study by Font and coll. [64] was done in a large cohort of 600 patients with SLE and reported only a modest variation in positivity to anti-dsDNA in presence (97%) or in absence (86%) of nephropathy. Recent studies in small patient cohorts confirmed the same inconsistencies [65,66]. In their review of the literature *ante* 2014 [44] Gensous and coll. reported heterogeneous results in terms of sensitivity (range from 27 to 100%) and specificity (range from 13 to 89%) of anti-dsDNA tests as predictors of LN producing therefore not conclusive results [67,68,69,70]. A pitfall in many studies was that incident and prevalent patients were not separated and the entity of therapies were not clearly reported that is a crucial point since potent immunosuppressive drugs, that are used in most aggressive forms of acute LN, certainly may modify the antibody levels.

A study reporting the simultaneous determination of anti-dsDNA, anti-nucleosome, and anti-histone (not considering different histones) in the largest cohorts of patients with LN and SLE ever reported, has been conducted in China and published in 2015 by Yang and coll. [71]. In their retrospective analysis, authors reported both prevalence and levels of anti-dsDNA in 921 patients with LN and in 778 non-renal SLE followed in a single center from 2002 and 2013. Anti-dsDNA were determined by a commercial kit (EUROLINE, Lubeck, Germany) and were found significantly positive in 63.3% of LN patients vs. 47.9% of SLE patients with a highly probable association with LN (OR 4.08, *p* < 0.001). Looking at levels, significant correlations were found between anti-dsDNA, anti-nucleosome, and anti-histone. What appears unusual in these results is the low positive percentage of anti-dsDNA in both categories of patients since it is unanimously believed that the positive percentage of anti-dsDNA in SLE is higher than 80%. We cannot exclude that ethnic differences may exist between Caucasian and Asian, a limit that reduces the impact of these findings.

An interesting association reported by few authors is between high anti-dsDNA levels and LN activity in terms of pathology indexes of proliferation and specifically with stage IV glomerulonephritis according to WHO [72,73]. Absolute antibody levels and rate of increment have been proposed as predictors of developing proliferative changes in glomeruli [66].

### 1.6. Anti-dsDNA Levels and Therapy

Changes in anti-dsDNA levels after therapies have been long since reported in clinical studies and is now an updated issue [45,74,75,76,77]. Anti-dsDNA antibodies represent, in fact, a significant part of a biomarkers panel utilized for the follow up of SLE patients. In general, a decrease of anti-dsDNA levels correlates with an increase of C3 levels and is associated with a low risk of lupus flare. Quite a number of therapies may induce a reduction of anti-dsDNA levels and prospective studies usually utilize this parameter as a secondary or surrogate outcome [74,75,76,77]. The same has been reported for renal flares [78]. Here the debate is about the frequency of tests since monthly sampling seems critical for an accurate determination of the association with disease activity [48,79]. A possible evolution of clinical value is to utilize changes in anti-dsDNA levels as an index of need of therapeutic intervention prior that clinical symptoms appear [80]. This possibility is now in discussion but it is clear that the possible benefit should be balanced with costs and needs a significant involvement of patient motivation.

### 1.7. Cross-Reaction of Anti-dsDNA with Neuronal Proteins

A final but important action of anti-dsDNA antibodies is their cross reaction with a specific aminoacid sequence present in neuronal *N*-methyl-*D*-aspartate (NMDA) glutamate receptor [81]. NMDA is an excitatory aminoacid released by the neurons and therefore the blockade of its receptor may produce neuronal damage and cognitive impairment. Kowal and coll. [82] showed that infusion of anti-dsDNA antibodies in mice cause hippocampal alteration; anti-NMDA receptors antibodies are also present in the brain tissue of patients with cerebral lupus thus supporting a cross-reactivity with anti-dsDNA. These observations suggest that anti-dsDNA may be connected with neurologic lupus and stimulate new studies on a clinical association with neurologic or psychiatric symptoms.

## 2. Autoantibodies to Nucleosome

The nucleosome is the basic dynamic element of chromatin that plays a strong impact in lupus autoimmunity. In the previous sections it has been already discussed how nucleosome and nucleosome components are variably exposed to the environment and which mechanisms should be involved in the formation of antibodies versus various antigenic sites (i.e., DNA, Histones, viral antigens, etc.). For anti-nucleosome antibodies, mechanisms are very complex and are outside the scope of this review; they have been described in details in a recent review by Rekvig and coll. [83] that contains also a critical reflection on their specificity and significance.

There is an increasing emphasis in considering anti-nucleosome antibodies in SLE diagnosis and complications. Association studies demonstrated that determining circulating anti-nucleosome antibody levels has higher sensitivity for lupus flares than other markers such as anti-dsDNA and can be detected in serum of SLE patients in strict concomitance with the abrupt onset of the disease [84,85,86]. Bizzaro and coll. [87] performed a systematic review of the literature up to 2012 with metanalysis that overall included 4300 patients. A pooled analysis of data showed a sensitivity of anti-nucleosome of 61% and a specificity of 94% for SLE. Anti-nucleosome were also detected in 25.3% of patients with mixed connective tissue disease (MTCD) and in 14.9% of systemic sclerosis [87]. Positive and negative likelihood ratio were 13.8 and 0.38 respectively indicating that positivity is mostly invariably associated with the disease. The results also showed a modest difference in diagnostic power compared to anti-dsDNA (sensitivity 59.9 vs. 52.4; specificity 94.9 vs. 94.2) [87]. Some authors have also proposed anti-nucleosome antibodies as markers of LN in small studies [85,88,89]. In the metanalysis performed by Bizzaro [87] the association of anti-nucleosome with LN did not, however, reach statistical significance. In the study by Yang [71], anti-nucleosome antibodies were detected in 59.8% of patients with LN and in 35% of non-renal SLE, a difference that was reported statistically significant *p* < 0.001. Based on the above results and, in spite of anti-nucleosome specificity and sensitivity are similar to anti-dsDNA, there is not any reason to double the laboratory approach to SLE and LN proposing the utilization of anti-nucleosome antibodies in clinical practice. There are also few weaknesses that hint, in some way, an applicative program of this test as already pointed out above: one is the high positivity of anti-nucleosome antibodies in patients with mixed connective tissue diseases (MCTD) and with systemic sclerosis [87]; a second flaw is about laboratory assays utilized for antibody detection [83]. One basic question is, in fact, the use of different antigens in ELISAs utilized for anti-nucleosome; one possibility is, in fact, to utilize the entire nucleosome prepared by digestion with micrococcal nuclease, a second is to use nucleosome stripped of H1. Associations studies are in favor of the second technique for its high specificity (i.e., 99.8% for H1-stripped nucleosome vs. 87% whole nucleosome) [87] but concerns exist about the composition of both. The possibility is that in different ELISAs, antibodies bind unspecifically histone variants that are not constant in nucleosome depending on the phase of a given cell.

In conclusion, anti-nucleosome antibodies give only modest advantages compared to anti-dsDNA for defining SLE and are also positive in other numerically important rheumatologic conditions such as MTCD and Sjogren syndrome. Moreover, there are a few technological flaws. With this in mind, the authors feel that it is not time to introduce an important modification in clinical practice at least until other studies can show clear difference in performance among tests for defining SLE.

## 3. Autoantibodies to Histones

Histones are cationic proteins that form the basic units of chromatin and form the backbone to which DNA is wrapped up. Of the 5 histone subtypes, H3 and H4 form the inner core of the nucleosome, H2A, H2B are in the outer portion and H1 is located outside the core in tight relationship with DNA. Histones have been recently investigated since they undergo several post-transductional modifications associated with apoptosis and netosis in SLE that may increase the immunogenicity [90,91,92,93]. Few studies have determined the circulating levels and specificity of anti-Histones in SLE; the largest study was recently performed by Yang and coll. [71] who determined serum levels of anti-histone antibodies without making any differentiation among H1, H2, and H3 and reported a lower percent of positivity (21.5%) in patients with SLE without nephropathy as compared to patients with LN (59.9%). Kiss and coll. [85] reported high serum levels of anti-histone (also in this case without any differentiation for major histone subtypes) in patients with LN compared to non-renal SLE. There is some concerns for the group of patients studied by these authors since they reported a very low positivity of anti-dsDNA (25%) that limits the interpretation of these data. Other authors showed specificity [94] limited to anti-H1 for SLE and for LN [72]. Overall, there is limited experience on anti-histones and specifically on antibodies to various histone species in SLE and LN to tempt any consideration.

## 4. Isotype Specificity of Nephritogenic Anti-dsDNA and Anti-Histone Antibodies

A main new issue in autoimmunity related to SLE and LN is the definition of specificity of autoantibodies correlated with the disease. In fact, as already discussed in the section on generation mechanisms of anti-dsDNA antibodies, it is possible that anti-dsDNA and anti-Histone form in response to infectious triggers; mononucleosis is an example [95]. Recent evolutions about the characterization of anti-dsDNA and anti-histone antibodies eluted from glomeruli of patient with LN demonstrated that IgG2 was the predominant isotype [56,57]; for their intra-glomerular localization, anti-dsDNA, anti-H2A, and anti-H3 IgG2 were defined as nephritogenic antibodies. Thanks to the same approach, it has been also shown that no anti-H1 antibodies can be eluted from kidneys demonstrating an improbable role in LN [56].

The finding of IgG2 specificity for nephritogenic antibodies opens to new considerations on the meaning of circulating levels prior and during the renal relapses in patients with SLE. In fact, the isotype of circulating anti-dsDNA and anti-histone had been found to be different in patients with SLE in relation to the presence and absence of renal flare and a shift from IgG1 to IgG2 was also observed prior and during the renal relapses. In spite this phenomenon has been documented in small cohorts of patients that were followed longitudinally [96]; it was considered of interest also because the same IgG2 isotype predominates for other potentially nephritogenic antibodies such as anti-C1q [97,98]. The characterization of IgG2 for antibodies eluted from the kidney and the prevalence of circulating anti-dsDNA and anti-Histones IgG2 in concomitance with the development of renal lesions suggest a direct pathogenetic role and points out that it is necessary a re-evaluation of circulating levels of isotype specific antibodies in LN.

## 5. Mechanisms of Isotype Specific Antibodies Generation

The mechanism for isotype switching to IgG2 has not been elucidated yet. Formation of anti-DNA antibodies requires that DNA is exposed outside the cell and is recognized by human memory B cells when presented by Toll-like receptors. There is experimental evidence that human memory B cells are directly stimulated by DNA deriving from neutrophils extracellular traps or NETs [99] that are produced by neutrophils in response to inflammatory or immunologic stimuli [100,101]. Circulating NETs are increased in patients with SLE [102]. NETs are mainly composed of DNA and are, in fact, a physical barrier for viruses, bacteria, and any other exogenous pathogens where they are entrapped and killed. In SLE, the complex of LL37-DNA that includes DNA deriving from NETs and the LL37 antimicrobial peptide, activates TLR9 [99] and stimulates the production of IgG2 [103,104]. Besides DNA, NETs contain several intracellular proteins that are post-translationally modified and become antigenic. Histones in NETs undergo acetylation that modifies their immunogenicity and makes them potential targets of lupus antibodies [92]. The mechanism of autoimmunity linked to post-translational modifications of proteins in NETs is not new and it has been widely investigated for generation of autoantibodies directed against citrullinated proteins in rheumatoid arthritis [105,106]. Of note, enzymatic citrullination of histones mediated by peptidyl arginine deiminase (PAD-4) has also been shown to be a specific marker of NETs and even necessary for NETs stability [107]. Overall, the mechanism of antibody formation linked to NETs can explain the IgG2 isotype specificity of antibodies eluted from the kidney and that are currently considered a category of antibodies linked with the pathogenesis of LN. Because of the presence of other antibodies in glomeruli that have been proposed by the co-authors with anti-dsDNA and anti-Histones in LN, it is not time to draw any conclusion. However, it is clear that this is a topic of interest that should be further expanded in the near future and experiments directly addressing the expression of isotype specific IgG2 antibodies by B-cells stimulated by DNA and proteins derived from NETs are needed.

## 6. Clinical Value of Coexisting Positivity to All Nucleosome Components

A few recent reports have pointed out that the simultaneous presence of more than one antibody of the autoimmunity panel may characterize complicated SLE. In the major three studies reporting the combination of antibodies, more than 2000 patients were analyzed [71,72,89]. The most interesting finding arising from this analysis was that the simultaneous positivity for anti-dsDNA and anti-Histone antibodies was associated with a greater risk of developing proliferative glomerulonephritis and that the rate of clinical remission was reduced in these patients. It is a common idea, shared by many investigators, that determining more circulating antibody (i.e., anti-dsDNA, anti-H2, and anti- H3) of those commonly utilized for monitoring the disease activity may add major benefits in the follow-up of SLE patients and enable to predict recurrence of the disease.

The predictive approach is now growing and the evolution is toward personalized medicine. New antibodies that go behind the nucleosome have become of interest and are the new frontier. They include antibodies to the complement and/or to organ specific antigens. Coexistence of more antibodies, here including those against the whole nucleosome components, represent the new perspective [57,108,109].

## 7. Conclusions

In conclusion, anti-dsDNA and more in general autoantibodies versus the whole nucleosome have been historically one of the most studied and popular topics in Medicine over the years. Many conclusions have been reached on the meaning of anti-dsDNA and anti-Histone autoantibodies and they now represent the cornerstone of any clinical diagnosis of SLE. The role of anti-DNA antibodies in complications of SLE, such as in LN, has also been clearly demonstrated although clinical studies have not defined their association with target organ complications. It is probably the time to consider that in SLE, other antibodies specific for different organs exist and that their role may be prominent over anti-DNA. Technology evolution in the broad field of antibody discovery would furnish new results in the midterm, thus new advances could appear and modify our current opinions.

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
