# Peer review of "An Update on Antibodies to Necleosome Components as Biomarkers of Sistemic Lupus Erythematosus and of Lupus Flares"

_ijms, 2019, doi:10.3390/ijms20225799_

Round 1

Reviewer 1 Report

The article by Ghiggeri et al reviews the current available data evaluating the role of various anti-nuclear antibodies in predicting disease activity in SLE patients. This is a thoughtful review that specifically addresses the relative roles of anti-dsDNA antibodies, anti-histone and anti-nucleosome antibodies in this respect, a topic of intense current debate.

ln 84-85: "discrepancies for individual patients and lack of repetitive results between different assays have been reported" - these are essentially the same thing. I think the authors were trying to point out that discrepancies can be seen for different methods used to identify the same type of autoantibody as well as among the same method but different kits and they should make this clear. There are several grammatical errors throughout the article that need to be addressed.

Author Response

We performed changes suggested by this Reviewer

Reviewer 2 Report

In this review, Gian Marco Ghiggeri and colleagues describe the association (mainly statistical link) of anti-dsDNA, anti-histone and anti-nucleosome antibodies with lupus flare and also with the occurrence of lupus nephritis. Although of interest this review could benefit from some discussion on B cells and therapies.
Indeed, I would appreciate whether the effect of different therapies on the expression level of these antibodies and the progression of the pathology should be discussed.
Also, the mechanism of B-cell activation/deregulation responsible for the selective production of IgG2 and IgG3 isotypes should be introduced.
Finally a paragraph to explain how IgG2 and IgG3 contribute to the pathogenesis is very interesting to define (receptor and targeted cells in lupus patients)

Minor points

Acronyms have to be defined (i.e., GBM).

Check for grammar errors (just take a look of the title)

Author Response

We have added in Mn2 two sections as requested by this Reviewer; one is about the effect of therapies on anti-dsDNA (pag 8) and the other is about mechanisms for IgG2 switching (pag 11-12). The first issue has been summarized looking at the most recent literature; the second issue has been discussed with some details and now I hope the content coincides with the Reviewer expectations.

Reviewer 3 Report

The review article by Ghiggeri et al. “Biomarkers of Systemic Lupus Erythematosus and of Lupus flares: An Update on Antibodies to Nucleosome Components” describes the prospective of anti-dsDNA and anti-Histone antibodies in diagnosis of SLE patients.

This article has several critical errors in Standard English grammar, spelling, comma etc. in addition to following major concerns that should be addressed before the manuscript is ready for publication….

Major critics:

Line2> Please correct the spelling errors in the title. Line28> It should be “updated review”. Line30> The phrase “with the objective” is unnecessary here, please remove it. Line31> It should be “serve as a stimulus”. Line34> Please correct the keyword “anti-ds DNA”. Line35> Check the font size for “anti-Histone abs”. Line 97> “anti-ds DNA are present they are powdered 6” ??. Line123> Reference 42 and 43 can be combined in in the same [ ]. Line127> “anti-ds DNS” needs to be corrected. Line175> Reference 71 can go with 68-70. Line182> “anti-da DNA” needs to be corrected. Line199> “therefore the bock of its receptor may produce” I assume authors want to say “blockade of its receptor”. Line210> It should be “The mechanisms involved in the formation”. Line252> “netosis” .The letter “n” not capital letter. There is inconsistency in writing of “coll.” “col.” for colleagues throughout this article. If data were presented in a tabular form while comparing among different studies would have been better in greater interest of this journal and readers?

Author Response

All errors outlined by this Reviewer has been modified

Round 2

Reviewer 2 Report

The authors addressed most of my concerns and I consider that this review should now be published in IJMS.
Please check in the manuscript because there are still some typos.

Author Response

editorial corrections have been done

Reviewer 3 Report

Although, the authors have addressed most of my concerns and this manuscript seemed much improved after the first review, I still have some minor corrections that need to be corrected.

Line 34> the word “an” should be removed.

Line 178> “Font and coll.”

Line 254> Replace with the word “hint”

Line 323> Replace with the word “enzymatic”

Line 329> should be “topic”

Line 330> Replace with “ in the near future”

Author Response

editorial corrections have been done